# A Clinical-Genetic Score for Predicting Weight Loss after Bariatric Surgery: The OBEGEN Study

**DOI:** 10.3390/jpm11101040

**Published:** 2021-10-17

**Authors:** Andreea Ciudin, Enzamaría Fidilio, Liliana Gutiérrez-Carrasquilla, Assumpta Caixàs, Núria Vilarrasa, Silvia Pellitero, Andreu Simó-Servat, Ramon Vilallonga, Amador Ruiz, Maricruz de la Fuente, Alexis Luna, Enric Sánchez, Mercedes Rigla, Cristina Hernández, Eduardo Salas, Rafael Simó, Albert Lecube

**Affiliations:** 1Endocrinology and Nutrition Department, Hospital Universitari Vall d’Hebron, 08035 Barcelona, Spain; aciudin@vhebron.net (A.C.); enzamaria.fidilio@vhir.org (E.F.); cristina.hernandez@vhir.org (C.H.); 2Diabetes and Metabolism Research Unit, Vall d’Hebron Institut de Recerca (VHIR), Universitat Autònoma de Barcelona, 08025 Barcelona, Spain; 3Centro de Investigación Biomédica en Red de Diabetes y Enfermedades Metabólicas Asociadas (CIBERDEM), Instituto de Salud Carlos III (ISCIII), 28029 Madrid, Spain; nuriag@bellvitgehospital.cat; 4Endocrinology and Nutrition Department, Hospital Universitari Arnau de Vilanova, Obesity, Diabetes and Metabolism Research Group (ODIM), Institut de Recerca Biomèdica de Lleida (IRBLleida), Universitat de Lleida, 25198 Lleida, Spain; liligutierrezc@gmail.com (L.G.-C.); kikke03@gmail.com (E.S.); 5Endocrinology and Nutrition Department, Hospital Universitari Parc Tauli, Medicine Department, Universitat Autònoma de Barcelona, Institut d’Investigació i Innovació Parc Taulí, 08208 Sabadell, Spain; acaixas@tauli.cat (A.C.); mrigla@tauli.cat (M.R.); 6Centro de Investigación Biomédica en Red de la Fisiopatología de la Obesidad y Nutrición (CIBEROBN), Instituto de Salud Carlos III (ISCIII), 28029 Madrid, Spain; 7Department of Endocrinology, Diabetes and Nutrition, Bellvitge University Hospital-IDIBELL, L’Hospitalet de Llobregat, 08907 Barcelona, Spain; andreusimoservat@gmail.com; 8Endocrine and Nutrition Department, Germans Trias i Pujol University Hospital, Germans Trias i Pujol Research Institute (IGTP), 08916 Badalona, Spain; spellitero.germanstrias@gencat.cat; 9Endocrine, Metabolic and Bariatric Unit, Department of General and Digestive Surgery, Vall d’Hebron University Hospital, Universitat Autònoma de Barcelona, Center of Excellence for the EAC-BS, 08035 Barcelona, Spain; vilallongapuy@gmail.com (R.V.); gordeju@gmail.com (A.R.); 10General and Digestive Surgery Department, Hospital Universitari Arnau de Vilanova, Institute de Recerca Biomèdica de Lleida, 25198 Lleida, Spain; mcruzdlfuente@hotmail.com; 11Surgery Department, Esofago-Gastric Surgery Section, Hospital Universitari Parc Taulí, Institut d’Investigació i Innovació Parc Taulí, 08208 Sabadell, Spain; aluna@tauli.cat; 12Scientific Department, Gendiag, 08028 Barcelona, Spain; eduardo.santos.salas@gmail.com

**Keywords:** obesity, bariatric surgery, weight loss, genetics, polygenic risk

## Abstract

Around 30% of the patients that undergo bariatric surgery (BS) do not reach an appropriate weight loss. The OBEGEN study aimed to assess the added value of genetic testing to clinical variables in predicting weight loss after BS. A multicenter, retrospective, longitudinal, and observational study including 416 patients who underwent BS was conducted (Clinical.Trials.gov- NCT02405949). 50 single nucleotide polymorphisms (SNPs) from 39 genes were examined. Receiver Operating Characteristic (ROC) curve analysis were used to calculate sensitivity and specificity. Satisfactory response to BS was defined as at nadir excess weight loss >50%. A good predictive model of response [area under ROC of 0.845 (95% CI 0.805–0.880), *p* < 0.001; sensitivity 90.1%, specificity 65.5%] was obtained by combining three clinical variables (age, type of surgery, presence diabetes) and nine SNPs located in ADIPOQ, MC4R, IL6, PPARG, INSIG2, CNR1, ELOVL6, PLIN1 and BDNF genes. This predictive model showed a significant higher area under ROC than the clinical score (*p* = 0.0186). The OBEGEN study shows the key role of combining clinical variables with genetic testing to increase the predictability of the weight loss response after BS. This finding will permit us to implement a personalized medicine which will be associated with a more cost-effective clinical practice.

## 1. Introduction

Obesity is a multifactorial and complex disease, caused by the contribution and interaction of environmental and genetic factors [1,2]. Its prevalence has increased dramatically in recent decades, and up to one-fifth of the whole world’s population is expected to live with obesity by 2025 [3]. The discouraging results provided by the conventional treatment have led to the progressive use of the bariatric surgery (BS) as a main treatment for morbidly obesity in Western countries [4]. 

However, around 30% of the patients that undergo BS do not reach an appropriate weight loss and/or do not resolve the comorbidities associated with obesity [5,6,7]. This fail is associated with a decline in health-related quality of life and patients report feelings of frustration, anger, and even depression [8,9]. Therefore, the identification of new predictive factors of response to BS seems mandatory [10,11,12,13]. This strategy will permit us to identify the best candidates to BS and even to select the type of BS technique, thus optimizing the health care resources. 

Inheritance is responsible for 40–75% of all the causes of obesity, a percentage modulated by the epigenetic influence [2,14]. Through genome-wide association studies (GWAS), a series of gene variants and single nucleotide polymorphisms (SNPs) in more than 120 genes have been linked with eating behavior, energy expenditure, response to diet, or lifestyle interventions [15,16]. As each variant alone has little effect on body weight, genetic predisposition is conditioned by the simultaneous presence of SNPs in multiple genes [11]. The possibility of elucidating the best combination of SNPs responsible for the variability of the response to BS in terms of weight loss offers the opportunity to design individualized therapy strategies. Our group has recently showed that an algorithm based on the selection of SNPs associated with predisposition to obesity, appetite regulation, and weight loss was able to predict the percentage of excess body weight loss (%EWL) after BS with high sensitivity and specificity [17]. It should be noted that this was a pilot study including only women who undergo a Roux-en-Y gastric bypass (RYGB), precluding the extrapolation of the results to the whole population.

The objective of the OBEGEN study is to confirm and extend our previous data regarding the added value of genetic testing to clinical variables in predicting the weight loss after BS. For this purpose, a total of 416 subjects (women and men) were evaluated, both RYGB and sleeve gastrectomy (SG) were included, and 50 SNPs were assessed. 

## 2. Materials and Methods

### 2.1. Statement on Ethics

The OBEGEN study was approved by the human ethics committee of the University Arnau de Vilanova Hospital of Lleida (CEIC-1743). All potential participants gave written informed consent to join the study, which was conducted according to the Helsinki Declaration and the Good Clinical Practice Guidelines. The study was registered in Clinical.Trials.gov- NCT02405949 (accessed on 16 October 2021). 

### 2.2. Study Design and Description of the Study Population

The OBEGEN project was a multicenter, retrospective, longitudinal, and observational study investigating the role of some genetic variants added to clinical variables to predict weight loss after BS. A total of 416 patients who underwent BS between January 2017 and August 2018 at the Obesity Units of four University Hospitals in Catalonia (Spain) were included. 

Eligible patients were men and women ≥18 years old, who underwent BS at least 18 months prior to the study. Among the 449 patients who met these criteria, 33 were excluded because of the following reasons: current pregnancy (*n* = 2), development of drug or alcohol abuse or eating disorders after bariatric surgery (*n* = 5), markedly mobility problems (*n* = 4), a different bariatric surgery technique apart from RYGB or SG (*n* = 9), use of weight lowering pharmacotherapy (*n* = 7) or a second surgical intervention required during follow-up (*n* = 6). The study flow chart is displayed in Figure 1. 

All the patients agreed to participate in the study and signed the informed consent underwent complete medical history, anthropometric measurement, physical examination, and DNA sampling.

### 2.3. Outcome Weight Measures

The primary endpoint was the %EWL at nadir. Excess body weight (EBW) was defined as the amount of weight that was in excess from the ideal body weight (IBW). IBW was estimated according to the 1983 Metropolitan Life Insurance Tables (use the midpoint for medium frame) [18]. The %EWL was calculated according to the formula: [total preoperative weight (kg) − weight after bariatric surgery (kg)/EBW (kg)] × 100. Those with a reduction in their %EWL >50% at nadir were considered “good responders” [19].

### 2.4. Genotyping

DNA was extracted from saliva samples and processed by GoldenGate^®^ Genotyping Assay for VeraCode. The genetic predisposition was assessed using the 50 SNPS in 39 genes included in a commercial nutrigenomic product, the Nutri inCode (NiC) (Ferrer inCode, Barcelona, Spain). This product includes SNPs that had previously been associated with susceptibility to weight loss, both in response to lifestyle intervention and BS [8,20,21]. In addition, Nutri inCode also includes selected variants of published GWAS studies or replication studies related to genetic susceptibility to regulate appetite and develop type 2 diabetes and obesity [22,23]. Finally, in the present study, the panel has been enriched with new 11 SNPs compared to our pilot study (Table 1) [17].

### 2.5. Statistical Methods

A normal distribution of the variables was established using the Kolmogorov–Smirnov test, and data are expressed as the mean ± SD, median (interquartile range), or as a percentage. Comparisons between groups were made using the Student’s *t*-test and the Mann–Whitney U test for quantitative variables, and the Pearson’s chi-squared for categorical variables. The relationship between continuous variables was examined by the Pearson linear correlation test. 

To assess the best predictive clinical–genetic risk score, patients were distributed into two groups according to the BS response: (i) %EWL ≤ 50% (*n* = 113); (ii) %EWL > 50% *n* = 301). On one side, the different SNPs were coded as 0, 1, or 2 according to the number of risk alleles associated with a favorable weight response. Clinical variables analyzed included gender, age at surgery, preoperative weight and BMI, type of surgery (RYGB or SG), EBW, and presence of type 2 diabetes. Univariate and multivariate logistic regressions were used to establish associations between the genetic and/or the clinical variables and the loss of weight. In this way, we generated three risk scores: (i) a clinical risk score, (ii) a genetic risk score, and (iii) the OBEGEN clinical-genetic risk score (OBEGEN-CGRS), which includes both the selected clinical and genetic variants in the multivariate logistic regression. 

Akaike Information Criterion (AIC)-based backward selection was used to remove not significant variables, from an initial model containing all the candidate predictors. The calibration of the logistic model’s adequacy was determined using the test of fit by the Hosmer–Lemeshow. The accuracy of different scores/models in discriminating those who obtained the objective weight loss (%EWL > 50%) from those who did not achieve the objective weight loss (for evaluating the prediction performance of the models) was evaluated using a Receiver Operating Characteristic (ROC) curve analysis. The cut-offs to calculate the sensitivity and specificity of the developed algorithms were selected as the point which maximizes the Youden index. An odds ratio with its 95% confidence interval was finally calculated. The total area under the ROC (AUROC) curve was interpreted following guidelines: 0.9–1.0 excellent, 0.8–0.9, good; 0.7–0.8, fair; 0.6–0.7, poor; and 0.5–0.6, not useful. Comparisons between the obtained AUROC were compared using the method of Hanley and McNeil.

All the contrasts were bilateral with a significance level of 0.05. The data were analyzed with the Statistical Package for the Social Sciences software (IBM SPSS Statistics for Windows, Version 20.0. Armonk, NY, USA).

## 3. Results

### 3.1. Baseline Characteristics of Patients

The main baseline characteristics of patients included in the OBEGEN study are shown in Table 2. After a follow-up period of 14.6 ± 0.8 months, 301 (72.3%) patients achieved a %EWL higher than 50%. Patients with a favorable weight response were younger, mainly women, and underwent RYGB.

### 3.2. Construction of a Clinical Risk Score

When only available clinical data were evaluated, the multivariable logistic regression model showed than age at BS, type of BS and presence of type 2 diabetes were independent risk factors for predicting a favorable weight loss in the entire population (Table 3). Therefore, a clinical risk score was developed including these three variables that showed an AUROC for predicting a good response to BS of 0.775 [95% confidence interval (CI) 0.731 to 0.814, *p* < 0.0001], with a sensitivity of 93.0% and a specificity of 50.4%. The calibration of the adequacy of the model determined by the Hosmer–Lemeshow test was 0.522.

### 3.3. Construction of a Genetic Risk Score

Additionally, when genetic data were analyzed alone, the multivariate logistic regression equation for predicting a favorable weight loss response after the BS included nine insSNPs located in ADIPOQ, MC4R, IL-6, PPARG, INSIG2, CNR1, ELOVL6, PLIN1, and BDNF (Table 3). This genetic risk score showed an AUROC of 0.648 (95% CI 0.597 to 0.696, *p* < 0.0001), with a sensitivity of 48.7% and a specificity of 75.0%. The calibration of the adequacy of the model determined by the Hosmer–Lemeshow test was 0.922.

### 3.4. Construction of the OBEGEN Clinical-Genetic Risk Score

Based on the clinical and genetic data from our population, we created the OBEGEN-CGRS, including age at surgery, type of surgery, presence of type 2 diabetes, and the nine SNPs associated with weight loss in response to BS (Table 3). The OBEGEN-CGRS score ranges from −4 to +4 points, with a cut-off point to define a good responder of 0.662. This predictive model showed an AUROC of 0.845 (95% CI 0.805 to 0.880, *p* < 0.001), with a sensitivity of 90.1% and a specificity of 65.5%. The calibration of the model’s adequacy determined by the Hosmer–Lemeshow test was 0.927. An internal validation of this clinical-genetic algorithm was performed using a Bootstrap method to quantify the uncertainty associated with the AUROC. The result was an AUC of 0.845 (95% CI: 0.800 to 0.888).

The OBEGEN-CGRS score showed a significant higher AUROC than either the clinical score (*p* = 0.0186) or the genetic score (*p* < 0.0001) (Figure 2).

## 4. Discussion

In this study we provide evidence that the combination of clinical plus genetic data is a reliable method to predict the weight response after BS. The OBEGEN-CGRS permits us to progress towards the personalization in the management of patients with severe obesity, seeking maximum efficiency with the least surgical damage.

Although BS provides successful weight loss in most of the cases, 25–30% of patients who undergo BS may not achieve the desired weight reduction [5,6,7]. This failure is considered multifactorial, with some preoperative factors associated with the hospital center (i.e., surgeons’ experience, bariatric procedure, preoperative education, and recommended weight loss before BS), the patient (age, gender, ethnicity, preoperative BMI, and comorbidities associated with obesity), and psychosocial features (economic resources, household type, and personality disorders) [11,12,13,24,25,26]. The real factors that predict weight loss following BS are still far to be determined. This is due to the inconsistency in reporting and the methodological weaknesses in analysis, which include a little if any consideration of genetic factors [21,27]. 

The development of new predictive tools for BS, based on some of the previous predictive factors, but at the same time fueled by new components, is a real need for clinicians who treat obesity worldwide. These instruments should help physicians to identify the best candidates who must undergo BS and to surgeons to optimize the surgical procedure.

Few studies have addressed the influence of genetics on long-term dynamic changes in body weight with ambiguous results [16,28]. The Swedish Obesity Study analyzed the impact of various SNPs from 11 genes in 1443 BS cases [29]. After the evaluation of 20 gene SNPs in 249 morbidly obese subjects undergoing RYGB, Velázquez-Fernández et al. showed that POMC rs1042571 was the only associated with favorable weight loss [30]. In another group of 1011 subjects, an increasing number of SNPs alleles in or near the FTO, INSIG2, MC4R, and PCSK1 genes negatively influenced weight loss trajectories after RYGB in those with an initial BMI >50 kg/m^2^ [10]. More recently, in a group of 146 individuals, carriers of another variant of the FTO gene (rs9939609) showed a lower success rate, as well as a greater and faster weight recovery beyond 2 years after BS [31]. Nevertheless, the same SNP was not associated with different weight loss after 6 months of SG in 74 morbidly obese patients [32]. Regarding the MC4R gene, among 1433 subjects with a follow-up period of 12 months after RYGB, carriers of the I251L variant lost 9% more weight compared with the noncarriers [20]. Finally, a prospective observational study with 105 patients evaluated SNPs in the leptin receptor, FTO and FABP2 genes [33]. This study showed that carriers of the LEP223 (rs1137101) experienced close to 25% lower excess weight at 12 and 24 months after bariatric surgery. 

Beyond the isolated study of one or the other gene, genetic risk scores composed by adiposity-related SNPs have been related with weight loss after RYGB or SG in Swiss, Danish, and Greek populations [34,35,36]. It should be noted that only three of the nine genes included in the OBEGEN-CGRS appeared previously reported in association with the weight loss response after BS [10,20,36]. This fact highlights the complexity of the genetic basis associated with the development of obesity, but also that the genes related to the therapeutic response may be different from those proposed so far. In the OBEGEN study, including both genders and different bariatric procedures, the combination of clinic and genetic data enhanced the predictive capacity of the genetic risk score, with improved sensitivity and specificity. Altogether, our findings support the use of genetic testing in clinical practice. 

Among the multiple variants of genes that were evaluated, nine of them interact with clinical variables to modify the weight response to BS in the OBEGEN study. While low serum adiponectin levels have been associated with central obesity, insulin resistance, metabolic syndrome, and type 2 diabetes, ADIPOQ (rs16861209) has been significantly associated with elevated fasting serum adiponectin levels [37,38]. Similarly, although IL-6 significantly increases the risk of obesity, in the PREDIMED trial carriers of the rs1800795 showed greater weight loss with the Mediterranean diet with supplements of olive oil compared to a Mediterranean diet low-fat diet than heterozygous and non-carrier carriers after 3 years of intervention [39,40]. In addition, PPARG (rs1801282) was associated not only with short-term (6-month) and long-term (2-year) weight loss but also with weight regain in the Diabetes Prevention Program [41]. PLIN1, a circadian lipid stabilizing protein in the adipocyte, has been associated with body weight regulation and PLIN1 (rs894160) with variability in weight loss [42,43]. Elovl6, a microsomal enzyme involved in the elongation of saturated and monounsaturated fatty acids with 12, 14, and 16 carbons, regulates mitochondrial function and thermogenic capacity in brown adipose tissue [44]. The BDNF (rs925946), INSIG2 (rs3771942) and CNR1 (rs6454674) variants are well stablished genetic determinants of obesity [45,46,47]. Similarly, MC4R (rs17782313) is associated with high dietary intake and different obesity-related phenotypic traits, such as insulin resistance, type 2 diabetes, and hypertriglyceridemia [48]. However, so far there are no data on the response to weight loss treatments regarding these last four variants.

The main differences between our CGRS and previous studies is located in the methodology assessment. Previous studies have been interested in analyzing the association between the number of genetic risk variables and the greater or lower weight loss following BS. However, we have focused our interest on the elaboration of predictive equations mixing clinical and genetic variables and the identification of predictive cut-off in those predictive equations. We have also calculated the prognostic capability of our score in identifying the “good” and “bad” responders to BS. We believe that this is a concept of great interest and more reliable in daily clinical practice. In this way, a patient who requires BS is exposed to an intervention that is difficult to reverse, so it is vital to be able to predict success or failure with a high degree of certainty. Currently, the choice of surgical technique is based on the baseline BMI and the presence of comorbidities. The introduction of a genetic predisposition score in the decision-making algorithms will bring us closer to the best selection of the patient, to choose the most convenient surgery, and to improve current health outcomes in our population.

There are some potential limitations that need to be considered when contemplating the results of our study. First, we evaluated a selected population of patients who underwent bariatric surgery, excluding those who underwent to other surgical techniques other than RYGB and SG, as well as those who had used weight lowering pharmacotherapy or had required a second surgical intervention. However, we believe that the inclusion of these more extreme cases might increase rather than reduce the reliability of our score. Second, the OBEGEN project is a retrospective one, meaning that no irrefutable clinical consequences can still be inferred to general population. Prospective long-term studies testing the genetic basis of patients with severe obesity before the recommendation of bariatric procedures are needed. Third, the characterization of favorable weight loss after BS is controversial [19]. In the OBEGEN study we have used a classic definition (%EWL>50) to better confirm our previous pilot study, so the results could be different if the chosen definition had been another. Finally, our model needs to be validated in an untrained dataset to fully demonstrate its applicability in the real clinical practice.

## 5. Conclusions

In conclusion, the OBEGEN project shows how the addition of genetic testing to the currently used clinical variables significantly improve our capability to identify patients with obesity who will be “good” or “bad” responders to BS in terms of weight loss. This information should help us both to personalize the therapeutic approach in severe obesity (e.g., select beforehand surgical techniques with a higher degree of malabsorption in those patients identified as “bad” responders), thus optimizing the limited health resources.

## Figures and Tables

**Figure 1 jpm-11-01040-f001:**
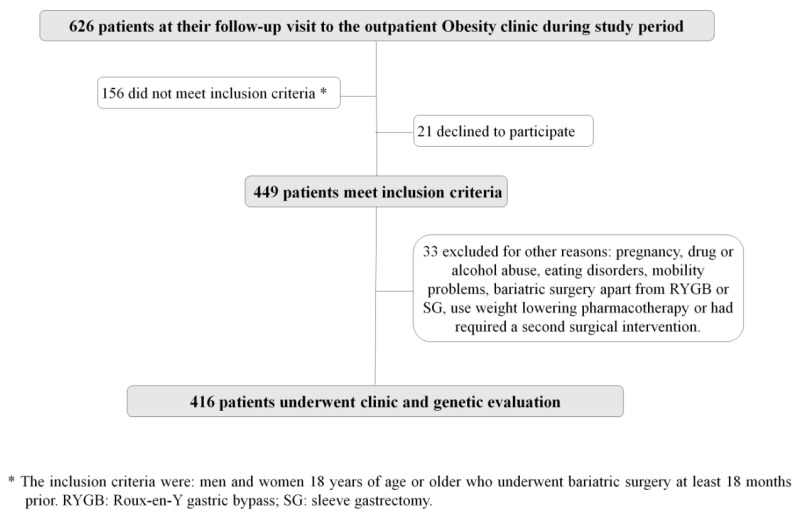
Flow chart of the OBEGEN study population.

**Figure 2 jpm-11-01040-f002:**
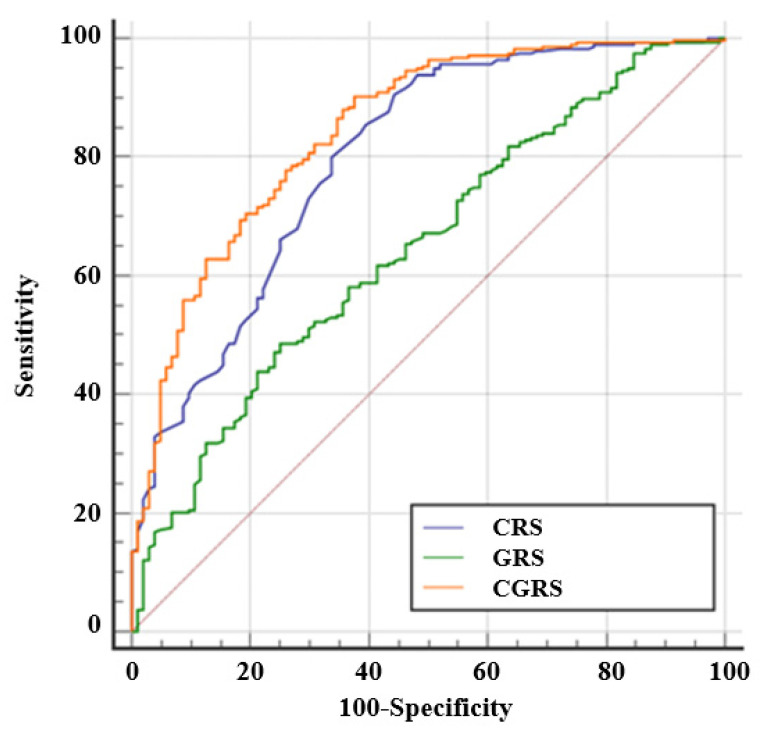
The predictive capacity of favorable weight loss (%EWL > 50%) obtained by the clinical risk score (CRS), the genetic risk score (GRS), and the clinical plus genetic risk score (CGRS) in our study population in the OBEGEN project.

**Table 1 jpm-11-01040-t001:** Selected genes and single nucleotide polymorphisms evaluated in the OBEGEN study.

Gene	Name	Chromosome Allocation of Human Ortholog	SNP
ACE	angiotensin I converting enzyme	17q23.3	rs4343
ADIPOQ	adiponectin, C1Q and collagen domain containing	3q27.3.	rs16861209 rs2241766
ADRB3	adrenoceptor beta 3	8p11.23.	rs4994 rs9693898
AGRP	Agouti related neuropeptide	16q22.1	rs11575892 *
AGTR1	angiotensin II receptor type 1	3q24.	rs5186
APOA2	apolipoprotein A2	11q23.3.	rs5082
APOA5	apolipoprotein A5	11q23.3.	rs651821
APOC3	apolipoprotein C3	11q23.3.	CD010
BDNF	brain derived neurotrophic factor	11p14.1.	rs6265 rs925946
CCDC93	coiled-coil domain containing 93	2q14.1	rs10490628 *
CDKAL1	CDK5 regulatory subunit associated protein 1 like 1	6p22.3.	rs7754840
CDKN2B	cyclin dependent kinase inhibitor 2B	9p21.3.	rs10811661
CLOCK	clock circadian regulator	4q12.	rs4580704 rs4864548
CNR1	cannabinoid receptor 1	6q15	rs6454674 *
ELOVL6	ELOVL fatty acid elongase 6	4q25.	rs682447
ESR1	estrogen receptor 1	6q25.1	rs3778099 *
FTO	fat mass and obesity associated	16q12.2.	rs9939609
GHRL	ghrelin and obestatin prepropeptide	3p25.3.	rs696217
IGF2	insulin like growth factor 2	11p15.5	rs680 *
INSIG2	insulin induced gene 2	2q14.1.	rs7566605 rs3771942 *
IL-1B	interleukin 1 beta	2q14.1	rs1143643 *
IL6	interleukin 6	7p15.3.	rs1800795
LEP	leptin	7q32.1.	rs12535708
LEPR	leptin receptor	1p31.3.	rs1137100
LPL	lipoprotein lipase	8p.22.	rs328
MC4R	melanocortin 4 receptor	18q21.32.	rs12970134 rs52820871 rs17700633 rs2229616 rs17782313
MTCH2	mitochondrial carrier 2	11p11.2.	rs10838738
NEGR1	neuronal growth regulator 1	1p31.1	rs2568958 *
PLIN1	perilipin 1	15q26.1.	rs1052700 rs894160
PPARA	peroxisome proliferator activated receptor alpha	22q13.31.	rs1800206
PPARG	peroxisome proliferator activated receptor gamma	3p25.2.	rs1801282
PCSK1	proprotein convertase subtilisin/kexin type 1	5q15.	rs6235
PON1	paraoxonase 1	7q21.3.	CD014
SIRT1	sirtuin 1	10q21.3	rs7069102 *
TCF7L2	transcription factor 7 like 2	10q25.2.	rs7903146
TMEM18	transmembrane protein 18	2p25.3	rs2867125 *
UCP1	uncoupling protein 1	4q31.2	rs45539933 *
UCP2	uncoupling protein 2	11q13.4.	rs659366
UCP3	uncoupling protein 3	11q13.4.	rs1800849
WFS1	wolframin ER transmembrane glycoprotein	4p16.1.	rs10010131

SNP: single nucleotide polymorphism. *: Genetic variants that have been added from the pilot study [17].

**Table 2 jpm-11-01040-t002:** Main clinical characteristics, metabolic, and anthropometry data of patients included in the study and according to the weight response to bariatric surgery.

	Total	%EWL > 50%	%EWL ≤ 50%	*p*
Patients, *n* (%)	416	301 (72.3)	115 (27.6)	<0.001
Female, *n* (%)	348 (83.6)	260 (86.3)	88 (76.5)	<0.001
Age (yrs)	48.3 ± 10.3	49.0 ± 10.4	51.5 ± 9.2	0.003
SG, n (%)	137 (32.9)	105 (34.8)	32 (27.8)	<0.001
RYGB, n (%)	280 (67.3)	218 (72.4)	62 (53.9)	<0.001
Initial BMI (Kg/m^2^)	44.3 ± 7.9	44.5 ± 9.6	44.2 ± 7.3	0.554
Initial weight (Kg)	113.0 ± 18.4	112.3 ± 20.3	113.4 ± 13.8	0.361
Excess weight (Kg)	49.3 ± 17.3	48.7 ± 19.2	51.3 ± 12.2	0.020
Nadir BMI (Kg/m^2^)	29.9 ± 5.8	28.5 ± 4.5	37.0 ± 6.8	<0.001
Type 2 diabetes, n (%)	173 (41.5)	128 (42.5)	45 (39.1)	<0.001
Hypertension (%)	286 (68.7)	204 (67.7)	82 (71.3)	<0.001
Dyslipidemia (%)	306 (73.5)	212 (70.4)	94 (81.7)	0.010
Sleep apnea (%)	116 (27.8)	88 (29.2)	28 (24.3)	<0.001

Data are mean ± SD, median (range) or n (percentage). SG: sleeve gastrectomy; RYGB: Roux-en-Y gastric bypass; BMI: body mass index. EWL: excess of weight loss. Hypertension was defined by increased systolic (≥140 mmHg) or increased diastolic (≥90 mmHg) blood pressure or using antihypertensive drugs, according to current guidelines. Dyslipidemia was defined using lipid-lowering drugs, decreased values of HDL cholesterol (men< 0.9 mmol/L, women <1.0 mmol/L) or by at least one increased value of total cholesterol (>5.2 mmol/L), LDL cholesterol or triglycerides (>1.7 mmol/L).

**Table 3 jpm-11-01040-t003:** Clinical and genetic variables that significantly predicts favorable weight loss after BS in the entire population of the OBEGEN project.

Clinical Variables		Genetic Variants		All selected Variants	
	Coefficient		Coefficient †		Coefficient †
Age	−0.03458	rs16861209	−0.30388	CRS	1.13897
Type of surgery	0.69588	rs17782313	0.32234	GRS	1.30048
Type 2 Diabetes	3.05077	rs1800795	−0.33407	Constant	−1.34401
Constant	−3.75577	rs1801282	0.33407		
		rs3771942	−0.17997		
		rs6454674	0.24788		
		rs682447	0.41113		
		rs894160	0.28848		
		rs925946	0.28604		
		Constant	−0.30768		

†: Coefficients in multiple logistic regression model. CRS: clinical risk score; GRS: genetic risk score.

## Data Availability

The data presented in this study are available on request from the corresponding author. The data are not publicly available due to privacy restrictions.

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
