# Peer review of "A Clinical-Genetic Score for Predicting Weight Loss after Bariatric Surgery: The OBEGEN Study"

_jpm, 2021, doi:10.3390/jpm11101040_

Round 1
Reviewer 1 Report
Here Andreea Ciudin and colleagues conducted a predictive modeling of weight loss after bariatric surgery in the forms of clinical-genetic score. It is an excellent work and worth publishing within Journal of Personalized Medicine readership. However, there are some minor concerns to be addressed before acceptance for publication.
There are many typos and reference format disruptions in Discussion section. The authors should recheck and revise accordingly.
The authors should add necessary details and discussions on how the 50 genes and single nucleotide polymorphisms evaluated in the OBEGEN study were chosen? By the way, these genes and SNPs are highly relevant to obesity and eating disorders.
Also, the predicting strongly show a favorable weight loss response after the BS in 9 SNPs located in ADIPOQ, MC4R, IL-6, PPARG, INSIG, CNR1, ELOVL6, PLIN1 and BDNF. The authors should discuss these genes and SNPs and their roles in obesity pandemics, like modulate appetite, overeating, thermogenesis, and metabolism.
Finally, it is highly recommended and possible that the gut microbiota may be a casual factor that induces the weight loss inconsistence after different surgery. Since the authors collected the saliva samples, maybe the authors also have the assess to the feces and future predictive modeling be developed on the mega-sequence of microbiome.
Author Response
REVIEWER#1
Thank you very much for the revision process of our manuscript entitled ““A clinical-genetic score for predicting weight loss after bariatric surgery: the OBEGEN study” (Manuscript jpm-1398904).
1.- There are many typos and reference format disruptions in Discussion section. The authors should recheck and revise accordingly.
These mistakes have been corrected. Thank you!
2.- The authors should add necessary details and discussions on how the 50 genes and single nucleotide polymorphisms evaluated in the OBEGEN study were chosen? By the way, these genes and SNPs are highly relevant to obesity and eating disorders.
Genetic predisposition was assessed using the genetic variants included in a commercial nutrigenomic product, the Nutri inCode (NiC) (Ferrer inCode). This product includes SNPs that had previously been associated with susceptibility to weight loss, both in response to lifestyle intervention and bariatric surgery. In addition, Nutri inCode also includes selected variants of published GWAS studies or replication studies related to genetic susceptibility to regulate appetite and develop type 2 diabetes and obesity. This information has been added in the new version of the manuscript.
3.- Also, the predicting strongly show a favorable weight loss response after the BS in 9 SNPs located in ADIPOQ, MC4R, IL-6, PPARG, INSIG, CNR1, ELOVL6, PLIN1 and BDNF. The authors should discuss these genes and SNPs and their roles in obesity pandemics, like modulate appetite, overeating, thermogenesis, and metabolism.
Following the indications of the reviewer, we have included information about the role of the nine SNPs that participates in the OBEGEN score in the new version of the manuscript
4.- Finally, it is highly recommended and possible that the gut microbiota may be a casual factor that induces the weight loss inconsistence after different surgery. Since the authors collected the saliva samples, maybe the authors also have the assess to the feces and future predictive modeling be developed on the mega-sequence of microbiome.
We thank the reviewer for this proposal, with which we fully agree. Unfortunately, we do not have data related to the gut microbiota to improve our results.
Reviewer 2 Report
Ciudin et al.’s investigation into genetic and clinical factors predicting weight loss surgery success is impactful and useful to the field. I believe that the results presented are promising, however more investigation is needed to solidify the results.
In its present format, the model is trained and tested on the same data-set, lacking validation. A key point of this study is highlighting increased predictability of BS success through the combination of genetic and clinical variables. Along those lines, the authors should show the application of their model on an un-trained dataset (i.e., one not known to the model before) to demonstrate real-world usability.
It would be useful to see whether responders vs non-responders can be clustered based on genetic data through, for example, a PCA plot.
Line 81, the authors reference their previous publication which utilized a similar methodology however there are some differences beyond what is mentioned here. For example, some genes (e.g., AGRP and SIRT1) were not included in their previous analysis. As the previous publication lends confidence to this analysis, I’d like to see the differences being highlighted either in the discussion or through a comparative table in the supplementary material.
Line 64-65, the authors state of unsuccessful outcomes to BS: “This has significant psychological effects, as patients feel like they have failed their last therapeutic option and experience frustration, anger, and even depression.” This suggests a causative link, however, in the absence of absolute proof, this would be better stated as a correlation, e.g. “Patients report feelings of frustration, anger and depression associated with failed BS.” Additionally, citations should be given.
In table 2, units should be given for initial weight and excess weight (e.g. kg).
Author Response
REVIEWER#2
Thank you very much for the revision process of our manuscript entitled ““A clinical-genetic score for predicting weight loss after bariatric surgery: the OBEGEN study” (Manuscript jpm-1398904).
1.- In its present format, the model is trained and tested on the same data-set, lacking validation. A key point of this study is highlighting increased predictability of BS success through the combination of genetic and clinical variables. Along those lines, the authors should show the application of their model on an un-trained dataset (i.e., one not known to the model before) to demonstrate real-world usability.
We fully agree with the reviewer that our model needs to be validated on an untrained data set to fully demonstrate its applicability in the real clinical practice. This fact has been pointed out as a limitation and a necessity of our study in the Discussion section.
2.- It would be useful to see whether responders vs non-responders can be clustered based on genetic data through, for example, a PCA plot.
Following the reviewer’s recommendation, we performed a principal component analysis (PCA) on clinical and genetic traits to provide the most information on traits that determine the response to weight loss after bariatric surgery. However, by using the PCA, the variables that retain almost all the relevant data that we previously identified by using the Akaike Information Criterion. In Figure 1, one genetic variable can explain 5% of the variable variation. However, when considered as a combination of genetic variants in a genetic risk score the capability of the genetic variants to explain the variable variation is much higher. The behavior of the clinical variables is different, as a single clinical variable can explain as much as the 24% of the variable variation.
Because genetic data is categorical, we respectfully consider Akaike’s information criterion to be a better method for evaluating our objective, comparing quite different models from each other. Therefore, this information has not been included in the new version of our manuscript.
3.- Line 81, the authors reference their previous publication which utilized a similar methodology however there are some differences beyond what is mentioned here. For example, some genes (e.g., AGRP and SIRT1) were not included in their previous analysis. As the previous publication lends confidence to this analysis, I’d like to see the differences being highlighted either in the discussion or through a comparative table in the supplementary material.
The reviewer is right, as we have enriched the panel with 11 new SNPs compared to our previous pilot study. These SNPs are in AGRP, CCDC93, CNR1, ESR1, IGF2, INSIG-2, IL-1B, NEGR1, SIRT1, TMEM18 and UCP1 genes. A comment has been added in the Materials and Methods section, and these variants have been highlighted in Table 1 for ease of comparison with our first and current work.
4.- Line 64-65, the authors state of unsuccessful outcomes to BS: “This has significant psychological effects, as patients feel like they have failed their last therapeutic option and experience frustration, anger, and even depression.” This suggests a causative link, however, in the absence of absolute proof, this would be better stated as a correlation, e.g. “Patients report feelings of frustration, anger and depression associated with failed BS.” Additionally, citations should be given.
We thank this comment to the reviewer. Therefore, we have rephrased this sentence and references have been added.
5.- In table 2, units should be given for initial weight and excess weight (e.g. kg).
We have introduced units in Table 2.

Round 2
Reviewer 2 Report
I am grateful for the authors response to my comments and believe they have addressed them satisfactorily. The changes made by the authors have improved the manuscript and where changes have not been made I feel that their response to my comments is sufficient.